# Tuning electronic and phononic states with hidden order in disordered crystals

**Nikolaj Roth** [1] ✉ **& Andrew L. Goodwin** [1]

Disorder in crystals is rarely random, and instead involves local correlations whose presence and nature are hidden from conventional crystallographic probes. This hidden order can sometimes be controlled, but its importance for physical properties of materials is not well understood. Using simple models for electronic and interatomic interactions, we show how crystals with identical average structures but different types of hidden order can have very different electronic and phononic band structures. Increasing the strength of local correlations within hidden-order states can open band gaps and tune mode (de)localisation—both mechanisms allowing for fundamental changes in physical properties without long-range symmetry breaking. Taken together, our results demonstrate how control over hidden order offers a new mechanism for tuning material properties, orthogonal to the conventional principles of (ordered) structure/property relationships.

The delocalised electronic and vibrational states key to many physical properties of periodic solids emerge from the collective behaviour of atoms and electrons on ordered lattices[1–3]. Random disorder breaks this emergence and drives localisation, resulting in scattering of electronic and vibrational states and lowering of electronic and thermal transport[4,5]. For strong random disorder, transport is completely stopped and—in the case of electronic properties—a metal-to-insulator transition can occur through Anderson localisation[6,7].

Disordered crystals present an interesting problem that, at face value, lies between these two extremes. Disorder is rarely random, and instead, many disordered crystals still obey strict local chemical rules that do not result in long-range symmetry breaking[8]. In this sense, such materials support a 'hidden order' that is not evident in conventional crystallographic analysis. A well-known example is the hydrogen-bonding network of water-ice $I_h$, where periodically arranged oxygen atoms each are covalently bonded to two of four nearby hydrogen atoms to give a non-periodic arrangement of $H_2O$ orientations[9]. Related states have been identified in mixed-anion perovskites[10,11], Coulomb-phase pyrochlores[12–15], and metal–organic frameworks[16]. An obvious and important question concerns the nature of collective electronic and/or phononic states in such systems: are they similar to those in ordered crystals or more closely related to those of amorphous solids? Or are they altogether different in character?

There are strong indications that hidden order may impact material properties. Short-range order in battery materials can influence ionic conductivities and charge-storage capacities by affecting the networks of mobile ions and vacancies[17,18]. Likewise, the nature of phonon broadening in disordered crystals has also been found to vary as a function of the type and extent of short-range order present[19–21] In the few systems known to exhibit hidden-order transitions—such as the magnetocaloric $Gd_3Ga_5O_{12}$[22]—the emergence of hidden order couples to thermodynamic anomalies. What remains entirely unclear is the nature of this link between hidden local order and collective phenomena.

Here we address precisely this problem by exploring the consequences of hidden order on the electronic and vibrational states of a model family of disordered crystals. The toy model we study is chosen because there is an obvious mechanism for varying the degree and nature of the hidden order it supports. We begin by introducing this model and explaining our approach for calculating electronic and phononic states for its various realisations. We then proceed to demonstrate a complex interplay between hidden order and the nature of collective states. In particular, we report three key findings: (i) that hidden order can be used to selectively broaden specific parts of the electronic or phononic band structure, (ii) that it modulates localisation in different ways, and (iii) that it can result in the opening of

[1]Inorganic Chemistry Laboratory, Department of Chemistry, University of Oxford, Oxford, UK. ✉e-mail: nikolajroth@gmail.com

band gaps without long-range symmetry breaking. We conclude by discussing generalisations of this toy model and the relevance of its behaviour to a range of physical systems.

## Results and discussion

### Hidden-order model

A useful toy model for exploring different types of correlated disorder is the two-dimensional system $A_2B$, where B atoms occupy a square lattice with A atoms positioned halfway between them (Fig. 1a), similar to the $H_2S$ layers in $H_3S$[23] or the $CuO_2$ layers in cuprate superconductors[24]. By introducing a distortion such that A atoms form one stronger and one weaker bond to neighbouring B atoms, several distinct types of disorder can be achieved. One possibility is for random distortions of A atoms, such as illustrated in Fig. 1b. In this case, the B atoms will have a varying number of strong and weak bonds. In many real systems, however, there will be local chemical rules that govern the types and geometries of bonds. One such example is for each B atom to have two strong and two weak bonds, which can be satisfied by a large number of configurations, with an example shown in Fig. 1c. These rules are similar to the two-in-two-out rule for hydrogen bonding in ice[9] and this square-lattice representation results in the well-known '6-vertex' statistical mechanical model[25]. Note that there are two types of B atom geometries, where the two strong bonds are either parallel or perpendicular to one another. A stronger chemical rule is then to have only perpendicular strong bonds, as illustrated in Fig. 1d, equivalent to the square-ice system[26]. In

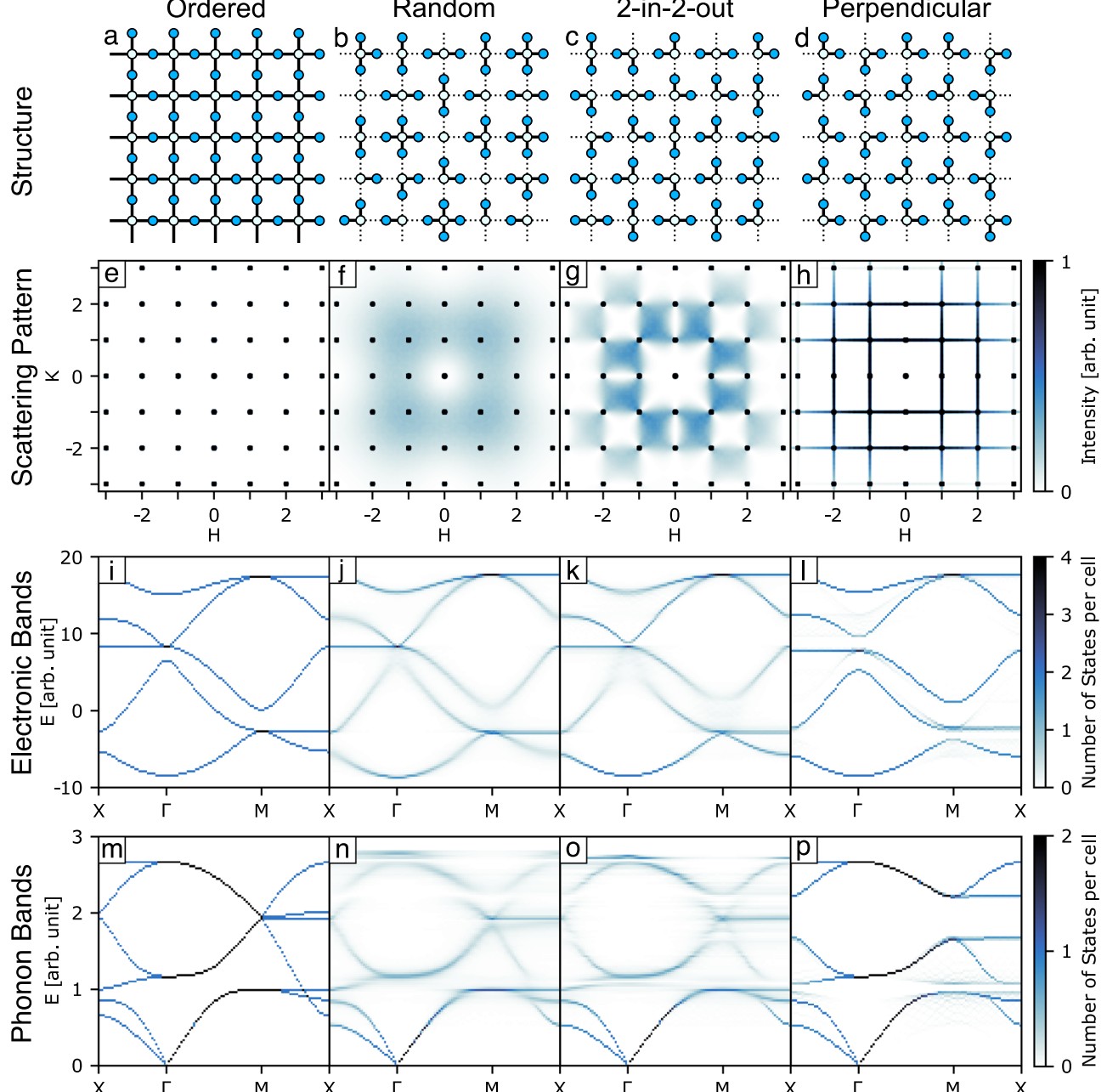

**Fig. 1 | The effect of random and correlated disorder on electronic and phonon bands. a** Ordered $A_2B$ structure with A halfway between B atoms on a square lattice. **b** A random configuration of A site distortions. **c** Two-in-two-out rule for correlated distortions. **d** Correlated disorder with perpendicular strong bonds. **e–h** The corresponding diffuse scattering patterns. The square lattice of black dots are Bragg peaks, which are several orders of magnitude stronger than diffuse scattering. **i–l** Electronic bands for these systems along special directions, with $\Gamma = (0, 0)$, $X = (^1/_2, 0)$ and $M = (^1/_2, ^1/_2)$. The energy scale is arbitrary and the zero point does not imply the Fermi level. **m–p** Phonon bands.

this highly constrained case, the strong A–B bonds are ordered in one-dimensional chains, but there is no three-dimensional bond order.

These distorted systems all have identical average crystal structures and therefore identical Bragg diffraction intensities. In this sense, the presence of additional local order is hidden from conventional crystallographic analysis. The clearest signature of this hidden order is through weak diffuse scattering. Figure 1e–h shows the single-crystal scattering pattern for each system. The square grid of black dots indicates the positions of Bragg peaks, which are several orders of magnitude stronger than the weak diffuse scattering lying between them. In the case of the undistorted parent structure (Fig. 1e) there is no diffuse scattering. Random distortions give broad diffuse scattering (Fig. 1f), while the two-in-two-out locally ordered system has characteristic structured diffuse scattering with pinch-points (Fig. 1g), reminiscent of those found in the scattering of three-dimensional (3D) spin-ice with a similar local rule[12,27]. Finally, the system with two strong perpendicular bonds has thin lines of diffuse scattering (Fig. 1h), indicative of long-range one-dimensional correlations.

## Collective electronic behaviour

We explore the effect of varying hidden order on the electronic properties of these models by calculating the electronic band structure using a semi-empirical tight-binding model with nearest neighbour hopping parameters. Drawing on the conceptual analogy to H and S arrangements in $H_3S$[23,28], we assign to B atoms a set of $s$, $p_x$ and $p_y$ orbitals but only a single $s$ orbital to the A atoms. On-site energies and hopping parameters for strong and weak bonds are modelled on the values calculated for $H_3S$, which has 2D layers with similar distortions of H between S on a square lattice[23,28]. Using this realistic parameter set allows some general effects to be illustrated. We note that the energy scale used is arbitrary and does not imply the Fermi energy lies at $E = 0$. Further details of our calculations are given in the methods section.

The electronic bands depend very strongly on the type and degree of hidden order. In the ordered state, the bands are well-defined in energy and disperse throughout the Brillouin zone with band crossings at the Γ and M points (Fig. 1i). Random distortions of the A sites change this picture, as shown in Fig. 1j. While the overall features and general dispersion are very similar to the ordered case, the bands are now broader. Hence, as anticipated, the electronic states are no longer well-defined in energy and will scatter as a consequence, reducing electronic transport.

By introducing the local two-in-two-out rule, significant differences to both the random and ordered cases are found (Fig. 1k). Now some of the bands have become narrower in energy again, while the gaps below the flat band at Γ and above the low-energy flat band at M have been filled with dilute states. Furthermore, the crossing above the flat band at Γ has lifted and given way to a small band gap. Changing the local order to the case of two perpendicular strong bonds per B site leads to very different effects, as shown in Fig. 1l. The bands are now generally narrow with states well-defined in energy, meaning electronic transport is not as hindered by scattering as in the two other disordered cases. In sharp contrast to the random and ordered systems, the bands crossing at Γ and M have now lifted and clear band gaps are observed. The dilute states filling some gaps in the two-in-two-out system are gone. There are also very weak additional band-like features between the strong narrow bands.

## Collective vibrational behaviour

We observe similar effects on the phonon spectrum as a consequence of correlations (Fig. 1m–p). In our calculations, phonon energies and eigenvectors are obtained by diagonalising the dynamical matrix using semi-empirical force constants between nearest neighbours. An arbitrary (but sensible) set of force constants was chosen to best illustrate the effects, as elaborated in the methods section. For reference, Fig. 1m

shows the phonon bands for the ordered system, where acoustic and optical phonons are well-defined in energy with crossing of four bands at the M point. Random distortions again give phonon bands similar to the ordered system but broadened in energy, resulting in increased phonon scattering (Fig. 1n). The broadening is least evident for the long-wavelength acoustic branches as these are most insensitive to variations in local configurations. The behaviour at the M point is now different: the four bands no longer cross as before, but change their dispersion to avoid the crossing.

The locally ordered two-in-two-out system has some differences from the random system in terms of the bandwidths (Fig. 1o), but it is the system with the strongest hidden order for which the phonon dispersion is most different (Fig. 1p). Here, the bands are almost all narrow, and a large band gap has opened throughout the Brillouin zone (Fig. 1p). Consequently, the type of correlations in disordered structures can also strongly impact properties that depend on vibrations, such as thermal transport. While some interplay between correlated disorder and phonon structure had been reported previously[19,20], a key result of this study is the demonstration that this interplay can be sufficiently strong as to open vibrational band gaps.

## Thermodynamic stabilisation of hidden order

In Fig. 2, we show the integrated electronic and phonon densities of states, which make clear that the band gaps seen along high-symmetry directions do indeed persist throughout the entire Brillouin zone. The emergence of band gaps for the two systems with strongest hidden order is conceptually important because, for the right filling fraction, a variation in hidden order type could lead to a metal–insulator transition. This would indeed be the case for this model system of $H_2S$-like layers, where the Fermi level falls within the gap as shown in Fig. 2a. Focusing on the emergence of electronic band gaps we note that the energies of the corresponding valence (low-energy) edge states are reduced in the 'perpendicular' hidden-order state relative to the ordered and random cases, which is also reflected in the total energy as shown in Fig. 2c. This stabilisation implies that the electronic energy of the system can be reduced through a concerted distortion to the hidden-order state. Such a transition is conceptually similar to a Peierls distortion, but is fundamentally different in that it proceeds without any global symmetry lowering. Ordered versions of the distortion can be produced with the same total system energy, but not lower, leading to the hidden-order version being entropically favoured, as discussed further in Supplementary Discussion 4. Coupling to strain, which is not considered in our model, may select an ordered ground state for a given system, but if the configurational entropy of the hidden-order arrangements is extensive, then any enthalpic driving-force for the order will be overcome at a finite temperature. Similar mechanisms may be at play in disordered 'orbital-molecule' states, such as in $LiRh_2O_4$ and $Li_2RuO_3$[29,30], where the structural distortions associated with valence electron localisation are local and not long-range ordered[31].

## Mode localisation

Correlations not only affect the form of the electronic and phonon band structure, but also change the delocalisation of modes. We show this in Fig. 3 by indicating the degree of delocalisation of electronic and phonon modes weighted by the number of states. Our key metric is a weighted participation ratio[32]: $D_\mathbf{k} = W_\mathbf{k} / \sum_i |c_i|^4$, with $W_\mathbf{k}$ the weight of each state at point $\mathbf{k}$, and $c_i$ the state coefficients, as elaborated further in the methods section. Taking each diagram in turn, we begin by noting that in the ordered system (Fig. 3a), electronic bands with dispersion are generally quite delocalised, while flat bands are localised. In the randomly distorted system (Fig. 3b), all bands have become more localised—as anticipated for disordered systems. The two-in-two-out rule gives rise to intermediate behaviour (Fig. 3c). But, most surprisingly, the most strongly correlated state (perpendicular strong

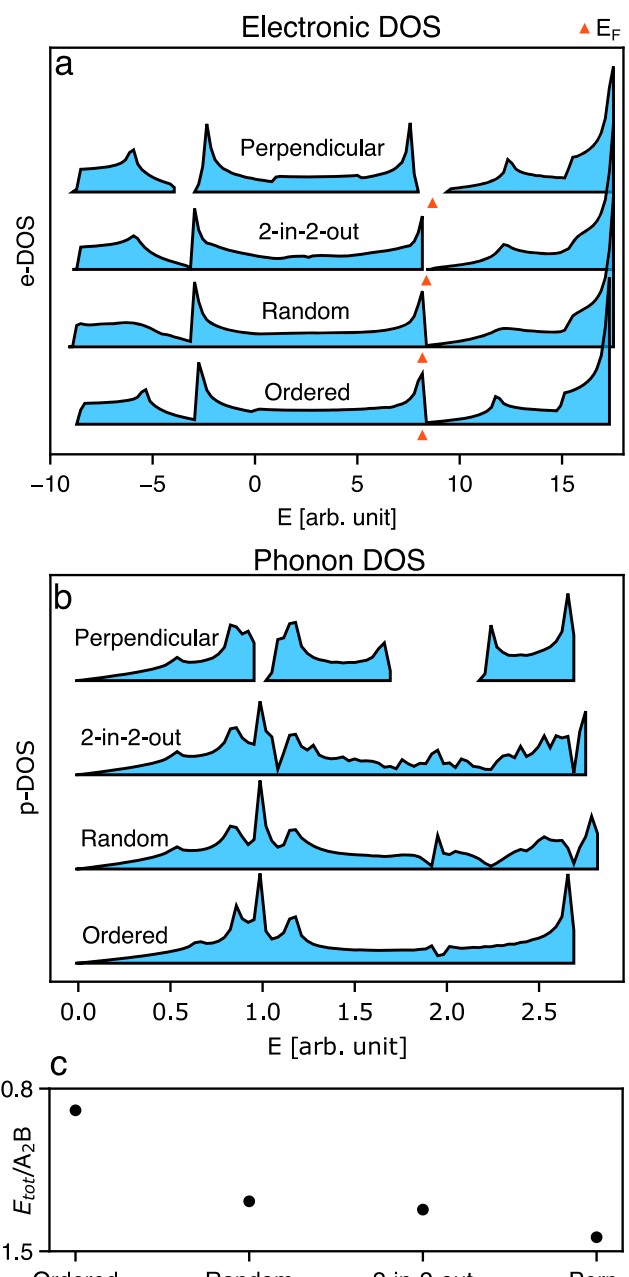

**Fig. 2 | Density of states (DOS) and total system energy. a** Electronic DOS. **b** Phononic DOS. Hidden order can induce band gaps that are not present in randomly disordered systems. Orange triangles show the Fermi level which fall right in this gap. The total electronic energy (**c**) shows how certain disorder types can be favoured.

highlight for interest here the unusual behaviour shown in Fig. 4h for the system with perpendicular strong bonds. This particular mode is completely delocalised with strong coherence along chains but mixing from chain to chain, causing chains to have phase shifts relative to each other. This is in contrast to the ordered, random and two-in-two-out systems, where the corresponding modes are all localised to a large extent. In a similar way, Fig. 4i–p shows the real-space representations of two types of phonon modes. Here colours indicate displacement direction–further highlighted by arrows–while saturation gives the corresponding amplitudes. The two vibrational modes illustrate how correlations can have different effects on modes, opening up the possibility of selectively (de)localising modes. These phonon modes are further discussed in Supplementary Discussion 2. To summarise, we find that hidden order not only affects the density and coherence of states but also their degree of delocalisation–often in quite nuanced and unexpected ways.

### Generality and extension to other systems

These results are not unique to the specific toy model on which we have focused but recur in other model systems containing hidden order. Figure 5a illustrates the case of valence-bond glass formation on the square lattice. Here we consider the electronic states formed through the overlap of $s$ orbitals for different distributions of neighbour-pair hopping parameters. A conventional, gapless, band structure emerges when hopping is uniform, but a gap opens when hopping is stronger between a site and exactly one of its four nearest neighbours ('Mixed' system in Fig. 5a). The limiting case corresponds to isolated dimer formation at half filling, which is conceptually related to the gapped states of valence-bond glasses such as $Ba_2YMoO_6$[33]. The low-energy branch corresponds to orbital combinations that are bonding with respect to individual dimers, but because the dimers are not periodically arranged, these combinations propagate with many different periodicities. The only 'forbidden' periodicity is M = $(\frac{1}{2}, \frac{1}{2})$, which corresponds to checkboard phase order and hence must always be antibonding with respect to dimers. Likewise, the high-energy branch corresponds to antibonding dimer combinations and is forbidden only at Γ, where all orbital contributions are in-phase and necessarily bonding. Figure 5b shows the corresponding density of states, illustrating the increasing band gap with a degree of dimerisation.

This same concept generalises to orbital-molecule formation in charge-disordered states. For example, at temperatures between 700 and 1100 K, the spinel $AlV_2O_4$ adopts a cubic structure with one crystallographically distinct vanadium site of formal charge $V^{2.5+}$. Pair distribution function measurements show, however, that the system contains a disordered distribution of spin-singlet $V_3^{9+}$ and $V_4^{8+}$ molecules 'hidden' within this average structure, rationalising why the material is not metallic in this regime[34]. In these various cases, as in that of the original $A_2B$ example explored above, the electron count associated with the formation of localised 'molecules' corresponds to the energy at which gap opening occurs. We expect that phonon band gap formation may be rationalised qualitatively in a similar vein, in that correlated disorder partitions phonons into inter- and intra-molecular contributions at low- and high-energy, respectively.

Hidden substitutional order can also result in gap opening. We illustrate this in the case of decorations of the triangular lattice, as shown in Fig. 5c. If an equal mixture of two components is distributed randomly across this lattice, then the electronic band structure of the crystal is substantially broadened at the Brillouin zone boundary. However, the formation of the so-called 'triangular Ising antiferromagnet' hidden-order state[35], in which triplets of mutually neighbouring sites always contain exactly two sites of one type, leads to gap opening at a filling fraction of one half (Fig. 5d). Hence, for the right electron count, a transition between random and correlated compositional disorder in this system could again result in a metal–insulator

bonds), gives delocalised states (Fig. 3d). Similar changes are observed for the phonon modes (Fig. 3e–h), for which the key difference is the resilience of delocalisation within the long-wavelength acoustic branches.

The variance in the degree of localisation is clearly exemplified by interrogating representative modes in real space. Figure 4a–h shows two examples of electronic modes for the different systems. The orbitals are coloured according to the wavefunction phase, while corresponding saturation is given by the wavefunction amplitude. Whenever modes are localised, atoms do not contribute equally to the wavefunction, which fragments into small coherent regions incoherent with respect to one another (see, e.g., Fig. 4b). We provide a detailed interpretation of these images in Supplementary Discussion 2, but

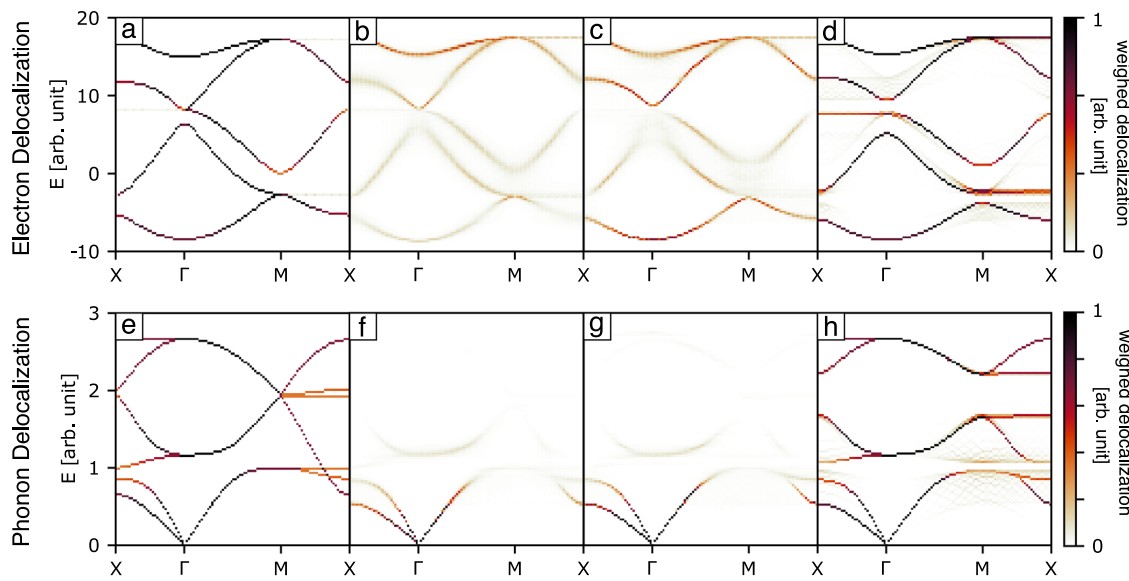

**Fig. 3 | Weighted delocalisation of modes. a–d** Delocalisation of electronic modes. **e–h** Delocalisation of phononic modes. The order corresponds to the order in Fig. 1.

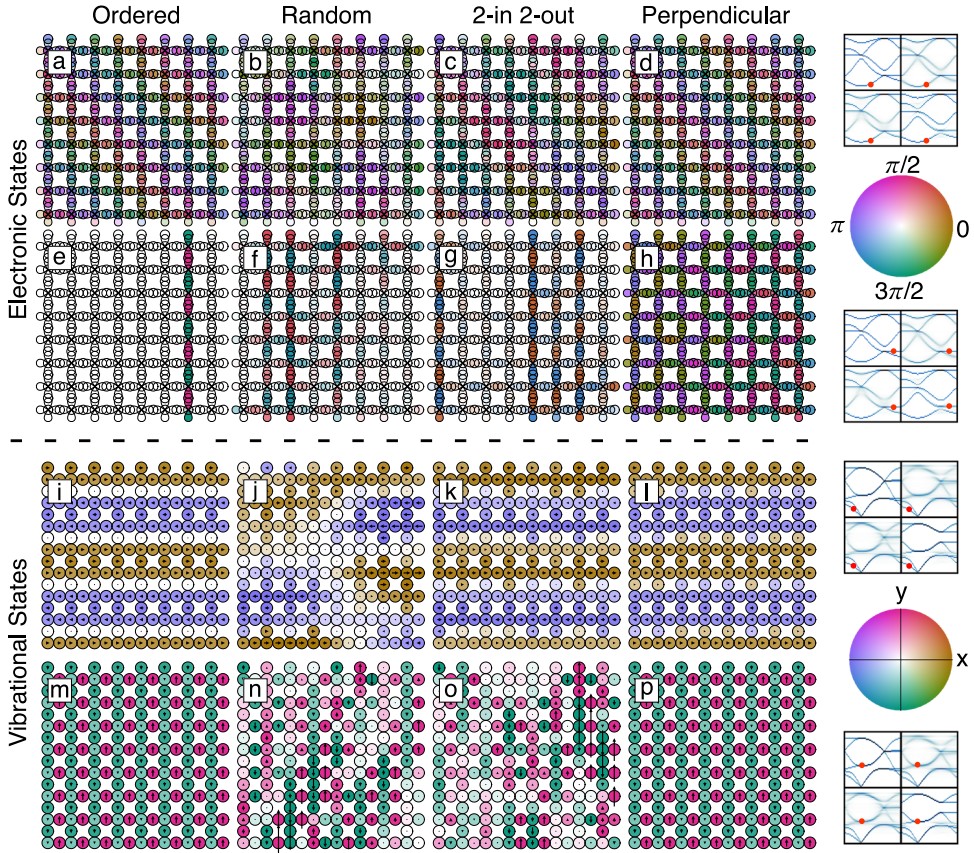

**Fig. 4 | Real-space view of modes. a–h** Two types of electronic modes for the different systems. Colour hue indicates the phase of the wavefunction, while saturation indicates the amplitude. **i–p** Two types of phonon modes for the different systems. Colour hue indicates direction of motion with saturation indicating amplitude. Arrows inside atoms further illustrate the movements.

transition without any long-range symmetry breaking. Any such transition need not be driven by electron–electron correlations, since these are excluded from our model; this point distinguishes the disorder-driven gap opening we observe here from Mott physics. This type of hidden order in the atomic distribution has been observed in the 2D layered semiconducting alloy $Re_{0.5}Nb_{0.5}S_2$, where it was demonstrated to affect the size of the band gap[36].

In Supplementary Discussion 5, we include a discussion of the further extension of our approach to 3D. The results are qualitatively the same as for two dimensions, albeit with some additional subtleties

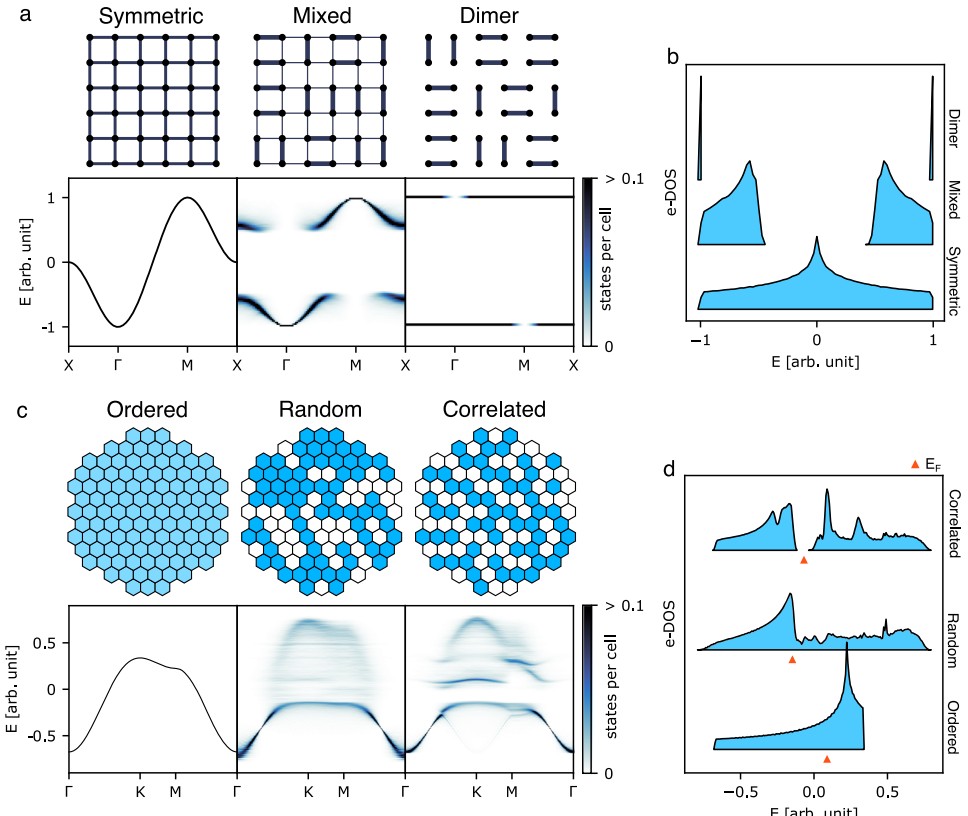

**Fig. 5 | Simple systems with band gaps induced by local order. a** Square system of $s$ orbitals with different types of hopping parameters forming partial and full dimers, and their corresponding electronic band structures. **b** Density of states for the square system. **c** Triangular system of $s$ orbitals from two types of atoms and corresponding electronic bands. **d** Density of states showing that half filling gives a Fermi level in the band gap induced by hidden order.

and avenues for control given the increased scope for geometric isomerism in 3D.

## Disorder engineering

Perhaps our key result has been to show clearly that correlations in disordered crystals have consequences for properties, as both electronic and vibrational modes are impacted in functionally important ways. Hence control over correlations offers a new handle with which to tune properties in functional materials. Moreover, because hidden order affects electronic and vibrational states in subtly different ways, it may prove possible to combine the effects of both to engineer functional materials with particularly desirable properties. We offer a handful of examples to demonstrate this point.

One topical family is that of thermoelectric materials, where the design brief is to combine a low thermal conductivity with large electrical conductivity in a gapped semiconductor, as captured by the phonon–glass–electron–crystal paradigm[37]. The conventional approach is to introduce disorder into a subset of atoms that do not contribute to electronic conductivity[38–40]. This is a design principle based on the idea of disorder being random and creating strong scattering of modes, which is why disorder on the substructure responsible for electronic conductivity is to be avoided. But our present study suggests an entirely new design strategy of introducing specific kinds of hidden order that at once broaden heat-carrying phonon modes whilst preserving narrow electronic modes in the conduction band. Additionally, one might even use correlated disorder to tune the electronic band gap so as to optimise thermoelectric performance[41]. In this context, we note that thermoelectric half-Heusler materials can be made with different local vacancy orderings but with identical average crystal structures and stoichiometries[42], indicating the possibility for tuning this class of materials through the concepts presented here.

The effect of disorder on topological insulators (TIs) is a problem of strong currency in the field of functional materials design. TIs are insulated in bulk but host conducting gapless edge or surface states. These gapless states are topologically protected and are robust against weak disorder[43,44]. For strong disorder, the non-trivial topological states can break down due to localisation. However, in some systems, strong disorder causes phase transitions from topologically trivial to non-trivial states, such as topological Anderson insulators[45,46] or disorder-induced topological Floquet insulators[47]. Quantised topological invariants are related to symmetries, but these can be broken into strongly disordered crystals. However, it has been shown that symmetry-stabilised topological invariants are still strictly quantised even in the presence of disorder that breaks symmetries locally yet restores them on average[48]. Since TI phases are robust to disorder, the disorder itself can be used to further engineer their band structures[49,50]. As we have shown here, the hidden order present within correlated disordered states can be used to control band structures, underlining the importance of understanding correlations in disordered TIs whilst also offering a new mechanism for tuning TI materials. The first 3D TI to be identified experimentally was the alloy $Bi_{1-x}Sb_x$[51], which supports disorder in the distribution of Sb and Bi. However, to the knowledge of the authors, the detailed nature of this disorder has not been studied. The compound is a TI for a range of $x$, and this might allow some degree of tuning of local order. Several such systems are known, such as $Bi_{1-x}Sb_xTe_3$[52] and $Mn(Bi_{1-x}Sb_x)_2Te_4$[53]. Other groups of TIs have been found experimentally to be disordered crystals[54,55]. Common to all these systems is that the nature of their disorder is not well understood, since earlier studies have only analysed average structures.

The same principles might be used to engineer band gaps and transport properties of photovoltaics, and—in principle—combining

effects of electronic and phonon band structures could tune electron–phonon coupling in superconductors. This could potentially be relevant to understanding materials like the topological super-conductor FeTe$_{1-x}$Se$_x$[56].

In an entirely different field, we anticipate that the link we demonstrate between hidden order and gap opening may have implications for the design of disordered photonic materials. The relatively recent demonstration of optical transparency in hyperuni-form structures has shown that subtleties of disordered networks can have fundamentally important effects on the optical band structure[57]. Likewise, control over the degree of short-range order has emerged as an unexpected design strategy for controlling visual appearance in photonic matter[58]. To the best of our knowledge, the concept of introducing hidden order within an otherwise-crystalline photonic medium as a means of introducing transparency has not yet explored, and may offer interesting new approaches for controlling matter–light interactions.

As a final point, we note that, because phases with different types of hidden order can have significantly different properties, it is more important than ever to develop experimental tools for probing hidden order in crystalline materials. The Bragg diffraction techniques used to determine crystal structures are sensitive only to long-range order, which is why it is often only the average structure of materials that is known. By contrast, diffuse scattering is sensitive to local correlations, but is several orders of magnitude weaker than Bragg scattering—this has limited its use historically[59]. The development of modern detectors and high-intensity x-ray, neutron and electron sources have now made it feasible to measure diffuse scattering much more routinely, allowing for the identification of distinct locally ordered phases[17,18,42].

## Methods

Electronic states are calculated from supercell configurations using a semi-empirical tight-binding model. Taking $\phi_i$ as the $i$th atomic orbital in the supercell, a basis of Bloch sums for wavevector $\mathbf{k}$ is

$$\Phi_{i\mathbf{k}} = \frac{1}{\sqrt{N}}\sum_{\mathbf{t}_m} e^{i\mathbf{k}(\mathbf{t}_m + \mathbf{v}_i)}\phi_i(\mathbf{r} - \mathbf{t}_m - \mathbf{v}_i), \tag{1}$$

where $\mathbf{t}_m$ is the position of the $m$th supercell origin, $\mathbf{v}_i$ is the position of the $i$th atomic orbital in the supercell, $N$ is the number of supercells, and $\mathbf{r}$ is the real-space coordinate vector. In the tight-binding approximation, the Hamiltonian then takes the form[60]:

$$H_{ij\mathbf{k}} = \sum_{\boldsymbol{\tau}} e^{i\mathbf{k}\boldsymbol{\tau}}\gamma_{ij\boldsymbol{\tau}} + \delta_{ij}E_{0i}. \tag{2}$$

Here, $\boldsymbol{\tau}$ are the vectors between atomic orbital $i$ and $j$ with nonzero matrix elements $\gamma_{ij\boldsymbol{\tau}} = \langle \phi_i(\mathbf{r})|H|\phi_j(\mathbf{r} - \boldsymbol{\tau})\rangle$, $\delta_{ij}$ is the Kronecker-delta and $E_{0i}$ the energy of orbital $i$ on an isolated atom. Here $\boldsymbol{\tau}$ is limited to nearest neighbours only and the matrix elements $\gamma_{ij\boldsymbol{\tau}}$ are given semi-empirical values for the different types of orbital combinations. In the present case, one type of atom is given one $s$ orbital and the other type one $s$ and a set of $p$ orbitals. The needed parameters in the present case are a set of four values comprised of $\gamma_{ss\sigma}$, and $|\gamma_{sp\sigma}|$ for short and long bonds, as well as parameters for $E_{0i}$. Directional dependence is taken into account using $\gamma_{sp\sigma} = l_x|\gamma_{sp\sigma}|$ for an s to $p_x$ element, where $l_x$ is the $x$-component of the normalised $\boldsymbol{\tau}$ vector, and similarly the s to $p_y$ and s to $p_z$ depend on $l_y$ and $l_z$, respectively. $p$ to $s$ orbital elements obey $\gamma_{ps\sigma} = -\gamma_{sp\sigma}$[60]. All other matrix elements are zero in this case. In other cases, more matrix elements would be needed, such as the $\gamma_{pp\sigma}$, $\gamma_{pp\pi}$.

Using a custom Python script, the Hamiltonian was constructed and diagonalised to obtain the eigenvectors and eigenvalues of the system at different $\mathbf{k}$. The bands were then unfolded to the Brillouin Zone of the primitive cell by calculating the weight of each state as[61]:

$$W_{\mathbf{k}} = \frac{1}{N_o}\sum_{o\in PC}\left(\sum_{i\in o} c^*_{i\mathbf{k}}\right)\left(\sum_{i\in o} c_{i\mathbf{k}}\right) \tag{3}$$

The sum $o \in PC$ are over the different orbitals of the primitive cell, and the sum $i \in o$ are those orbitals in the supercell which are equivalent in the primitive cell. $c_{i\mathbf{k}}$ are the coefficients of the normalised eigenvectors in the Bloch sum basis and $N_o$ the number of orbitals in the system. The number of states per cell for each mode is then given as $2N_{0\in PC}W_{\mathbf{k}}$, where $N_{0\in PC}$ is the number of orbitals in the primitive cell and the factor of two takes into account the spin degree of freedom. The weighed degree of delocalisation of each mode, $D_{\mathbf{k}}$ was calculated as[32]: $D_{\mathbf{k}} = W_{\mathbf{k}}/\sum_i|c_i|^4$.

Phonon modes were calculated in a very similar way by con-structing and diagonalising the mass-adjusted dynamical matrix from the eigenvalue equation:

$$DU = \omega^2 U. \tag{4}$$

Here $D$ is the mass-adjusted dynamical matrix, $U$ is the eigenvector of mass-adjusted elementary movements and $\omega$ the energy. The method for phonon calculations follows that given in detail in ref. 62. Elements of $D$ are given by

$$D_{ij\mathbf{k}} = \frac{1}{\sqrt{m_{ai}m_{aj}}}\sum_{\boldsymbol{\tau}} e^{i\mathbf{k}\boldsymbol{\tau}}K_{ij\boldsymbol{\tau}}, \tag{5}$$

where $i$ and $j$ now reference the elementary movements of all atoms in the supercell along cartesian axes. $m_{ai}$ is the mass of the atom to which the $i$th elementary movement belongs. $K_{ij\boldsymbol{\tau}}$ is the force constant between elementary atomic movements $i$ and $j$. The diagonal elements $D_{ii\mathbf{k}}$ need to conserve force balance: $D_{ii\mathbf{k}} = -1/m_{ai}\sum_{j\neq i}K_{ij}$. Again, only the nearest neighbours are included. Two types of force constants are used: $K_\perp$ and $K_\parallel$ for perpendicular and parallel movements of nearest neighbours, with two possibilities for short and long bonds for each.

The phonon bands were unfolded in the same way as the elec-tronic bands using

$$W_{\mathbf{k}} = \frac{1}{N_u}\sum_{u\in PC}\left(\sum_{i\in u} c^*_{i\mathbf{k}}\right)\left(\sum_{i\in u} c_{i\mathbf{k}}\right), \tag{6}$$

where $u$ are the elementary displacements in the primitive cell, $N_u$ the number of elementary displacements in the supercell and $c_{i\mathbf{k}}$ the coefficients of the normalised eigenvectors of $D$. The weighed delo-calisation was then calculated in the same way as for the electronic modes.

The electronic and phononic band structures were calculated on configurations with 32 by 32 atoms and averaged over 30 different configurations. For the electronic bands, values were chosen to be close to those calculated for H$_3$S, so as to keep them realistic. This was done using the minimal tight-binding model from ref. 28, where values for orbital energies are taken relative to the sulphur $s$ level, with $E_{0Ss} = 0$, $E_{0Sp} = 8.16$ eV and $E_{0Hs} = 6.42$ eV. The hopping elements used for the 2D simulation were rounded to the nearest integer values, $\gamma_{ss\sigma} = -5$ and $-3$ eV for strong and weak bonds, respectively. Similarly, $|\gamma_{sp\sigma}| = 6$ and $4$ eV were used. For the calculation of the density of states, we averaged over 100 configurations of size 60 by 60 cells. For the 3D systems shown in the SI, the exact values for H$_3$S given in ref. 28 were used for configurations with 16 atoms along each dimension and averaged over 10 configurations. These are $\gamma_{ss\sigma} = -4.69$ and $-2.98$ eV and $|\gamma_{sp\sigma}| = 5.69$ and $4.3$ eV. For the phonon calculations, parameters were chosen to give clear band structures. In the 2D systems, the masses for the two types of atom were $m_A = 0.8$ and $m_B = 1$. Values for the force constant were chosen as $K_\parallel = -2$ and $-1$ for strong and weak

bonds, respectively, as well as $K_\perp = -0.6$ and $-0.2$. For the 3D systems presented in the SI, values used are $m_A = 0.75$, $m_B = 1$, $K_\parallel = -2$ and $-1$, and $K_\perp = -0.4$ and $-0.2$. In general, the averaged values for strong and weak bonds were used for the calculation of the ordered system. The Diffuse scattering intensity was calculated using the Scatty software[63], using configurations with 60 by 60 atoms and averaged over 100 different configurations.

For the dimer model simulations shown in Fig. 5a, b, configurations of size 32 by 32 atoms were used for the band structures while 80 by 80 atom configurations were used for the density of states calculations. Each position on the lattice was given an $s$ orbital with an energy of 0. Hopping elements were chosen such that the average was conserved. For the symmetric model, all hopping elements $\gamma_{ss\sigma} = -0.25$ were used. For the mixed case, values of $-0.7$ and $-0.1$ were used for strong and weak bonds, respectively. For the dimer state, values of $-1$ and $0$ were used.

For the hexagonal system shown in Fig. 5c, d, configurations of size 36 by 36 atoms were used for band structures, and 80 by 80 atoms were used for the density of states calculations. The two types of sites (white and blue) were given $s$ orbital energies of $-0.25$ and $0.25$, respectively. Hopping elements were chosen as $-0.05$, $-0.1$ and $-0.2$ for white-white, blue-white/white-blue and blue-blue hoppings, respectively. For the ordered system, the averaged values were used for both energy and hoppings.

## Data availability

The data that support the findings of this study are available from the corresponding author upon request.

## Code availability

The custom python code used in this study is available from the corresponding author upon request.

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

## Acknowledgements

N.R. acknowledges the Independent Research Fund Denmark (DFF) for funding through the International Postdoctoral grant 1025-00016B. A.L.G. thanks A. R. Overy (Oxford), M. G. Tucker (SNS), A. Simonov (ETH Zurich), and C. Romao (ETH Zurich) for discussions, and the European Research Council for funding (Grant 788144).

## Author contributions

N.R. developed the concept, carried out the computational work, analysed results and prepared the figures. A.L.G. provided suggestions for analysis and contributed to the interpretation of results. N.R. and A.L.G. wrote the manuscript together.

## Competing interests

The authors declare no competing interests.
