## [Peer Review File · Nature Communications]

REVIEWER COMMENTS

Reviewer #1 (Remarks to the Author):

In this manuscript, the authors discuss the effects of correlated structural disorder on the spectra of electronic and phononic excitations. They argue that correlated disorder that is "hidden" from gross crystallographic probes (but not from, e.g., diffuse scattering measurements) can lead to qualitatively distinct outcomes from random structural disorder. In particular, they single out the opening of spectral gaps as well as effects in the localization of modes. Finally, they connect some of these ideas to properties of interest for applications, including thermoelectrics, topological insulators and photovoltaic materials.

Overall my impression of this manuscript is mixed. The paper is very well written, with the core ideas explained in a completely lucid manner. I do not doubt the technical correctness of the results, and the associated figures do an excellent job communicating the authors' key points. However, the core idea itself seems underdeveloped. The question the authors are addressing is how correlated disorder is different from uncorrelated disorder with respect to electronic and vibrational states. While simulation results for minimal models are presented, only a superficial interpretation of those results is presented. For example:

- Why does correlated disorder appear to open a band gap? Is there any conceptual understanding of this key result?
- Why does it do so at some high-symmetry points but not others? [e.g. in panels j,k of Fig. 1 a gap opens at Gamma, but at M. In panels n,o a large gap opens at high energy at M, but not at low energy and not at all at Gamma].
- Why is the gap opening locations different for phonons vs. electrons?

The lack of any further detail or exploration of the obtained results makes it difficult to ascertain their significance. For example, the 2-in/2-out case should have algebraic correlations between the atomic positions, while other kinds of correlated disorder may only yield exponential decaying correlations, and the "perpendicular" state has long-range correlations along each chain. Are the behavior of these correlations important to the result? Given that only a few simplified models are explored in this work, it is unclear to me how general the authors' result really is, given the absence of a conceptual framework.

This issue I think is my main concern with the work. While the presentation is excellent and the question posed is interesting, without a more conceptual understanding it isn't clear to me that this work should be published in its current form. My recommendation would thus be to not accept the manuscript in its current form, but perhaps to reconsider if the authors could provide a more conceptual picture for their core results.

More technically, I have a few questions/comments about more specific statements:

1) On page 7 the authors discuss thermodynamic stabilization of hidden order. They argue that the opening of a gap in the phonon or electron DOS for the "perpendicular" order suggests that a kind of Peierls distortion type transition could occur, favoring the "hidden order" state. While interesting, a natural question would be whether reducing the degeneracy further would result in an even larger gap. I.e. selecting a particular state from the manifold of "perpendicular" states could give rise to an even lower energy. If so, it is not clear to be that the partial favoring discussed by the authors is meaningful.

2) While the spectral plots shown in Figs. 1,3 seem satisfactorily converged, the DOS presented in Fig. 2 remains very noisy. In the supplemental information, the authors state 30 random samples were used. This is not nearly enough to obtain a converged DOS, as can be seen in Fig. 2. From this figure the difference between the random and correlated disorder cases seems well within the noise and thus it isn't clear what kind of conclusions can really be drawn.

3) When discussing the degree of localization of the modes, can the metric used distinguish between 0D and 1D localization? Given the "chain" like structure of the perpendicular states, it

seems possible they could be localized in one direction but not the other. I'd imagine this could be diagnosed by looking at scaling in system size. Can the authors comment on this?

Reviewer #2 (Remarks to the Author):

This manuscript describes a theoretical demonstration of the influence of correlated disorder on electronic and phononic properties. The paper is well written, and the results are clear and interesting.

I have one quite general criticism and then a number of specific comments that the authors should consider. Once addressed, I feel this manuscript would be appropriate for publication in Nature Communications.

My general criticism is that the authors refer to the correlated disorder that they model and analyse, as "hidden order". I can understand why they do this, as such incomplete order does not show up in conventional diffraction studies, and they make the case that it is often overlooked (which I am sure is the case). However, it isn't really "hidden". As the authors show, the different manifestations of correlated disorder show quite distinct diffuse scattering. While this is a criticism, I would be fine with it. However, "hidden order" is a well-used and well known term in correlated electron physics, and typically refers to thermodynamic phase transitions which have no easily-interpretable signature in structural probes, such as neutron and x-ray scattering. The best appreciated specific example is that of the heavy fermion superconductor URu₂Si₂, which the authors refer to, and which displays a very strong thermodynamic anomaly at ~ 17 K in zero field and ambient pressure. There have been many unsuccessful attempts to identify weak Bragg peaks associated with this thermodynamic anomaly, and hence the state below ~ 17 K is referred to as a "hidden order" state. There have been many proposals and speculations as to the nature of this "hidden order" state, but all the ones which I am familiar with propose exotic electronic ordering, such as high order multipolar order consistent with 5f-electron physics, and none propose a structural correlated disorder origin.

So, at least some of the community will be confused by the use of "hidden order" to describe the correlated disorder that is the subject of this manuscript. That is unless the authors are suggesting that correlated disorder is relevant for URu₂Si₂. Were that the case, among other things, the authors would need to discuss why a thermodynamic phase transition is required.

For this reason, I strongly recommend that the authors either simply refer to the "hidden order" as correlated disorder, or provide another descriptive term, or better explain the relationship between the "hidden order" described here and that commonly used in the correlated electron community.

My more minor comments are:

1- A very minor point: when I look at the 2D structures in Fig. 1 a) – d) I immediately think of CuO₂ planes, as occurs in many quasi-2D high temperature superconductors. Would this be inappropriate? I imagine H₃S is as good or perhaps a better conceptual analogy to use.

2- As the calculated electron and phonon bands span the 2D Brillouin zone shown in Fig. 1 and 3, along high symmetry directions involving the X and M points. It may be useful to provide a drawing of the 2D Brillouin zone indicating these points, so that a less expert reader could follow this more easily.

3- The measures of electron and phonon delocalization shown in Fig. 3 are interesting, however, how these measures are determined and presumably related to the electronic and phononic band structures shown in Fig. 1 requires that the reader continue into the Methods section. Given the prominence of this result – accounting for 1 of the 4 figures in the manuscript, I would prefer to have these details discussed, at least qualitatively, where Fig. 3 itself.

Reviewer #3 (Remarks to the Author):

This paper presents simulations that demonstrate how varying local order types within a toy model average crystal structure can give rise to different features within electronic and phononic band structures, specifically blurring of bands and opening of new gaps. The work is from a leading group in the field of hidden order within crystals and the calculations appear technically sound. Graphical comparison of results throughout is excellent.

However, glossing over the electron-filling aspect of the electronic band structure is a weakness. Even in a toy model, electronic structural stability is implicit for the pDOS to be meaningful and so some number of electrons (such as for H₂O) should be used to define a proper Fermi level. The eDOS's for the ordered and random models in Fig. 2a look rather similar with the implication that both would be metallic for EF somewhere in the middle of the band which seems odd. The eDOSs should also be sensitive to the magnitude of displacement of A from the midway position used in the calculations. Was this varied and does it have any significant effect? The issue of electron-electron correlation which opens Mott type gaps in many materials is also not mentioned - in real systems it may be hard to distinguish this from disorder-induced gaps.

The main novelty of the paper seems to be as an educational exercise in being able to eyeball how the different types of real space disorder translate into momentum space features of diffuse scatter and band structure changes. As such I found it very interesting and I think it will have impact on others working on disordered materials.

The downside is that I'm not sure if any of the calculated features will be observable. People have been studying hidden disorder materials such as those refs 9-24 and also using supercell models to simulate band structures of disordered materials for many years but I do not recall seeing specific examples where a disorder model is linked to specific changes in electron/phonon structure. There is the long-known link of disorder to metal-insulator transitions, going back to Anderson localisation theory, but I'm not sure the present work adds much to this, perhaps because electron-electron correlation is neglected as above. Fig. 2a,b, shows some large gaps being opened in some models - are these reported for the corresponding materials or why were they not found?

The Concluding Remarks give some suggested connections to topical materials like topological insulators, but there is a lack of specific predictions. How would I use this paper to do some new disorder calculation or study that I couldn't already do if I was working on TI's or other materials?

Overall I find this a stimulating paper that could be accepted for publication after consideration of the above comments.

RESPONSE TO REVIEWERS' COMMENTS

We would like to thank the reviewers for their insightful comments and suggestions. In light of these reviews, we have now carried out a series of new additional calculations, and have made a large number of changes to the paper. These changes include, amongst other things, a new results section, a new figure (Fig 5) and a new section in the supporting information. We provide below a point-by-point response to all the various criticisms and questions raised.

Reviewer 1

In this manuscript, the authors discuss the effects of correlated structural disorder on the spectra of electronic and phononic excitations. They argue that correlated disorder that is "hidden" from gross crystallographic probes (but not from, e.g., diffuse scattering measurements) can lead to qualitatively distinct outcomes from random structural disorder. In particular, they single out the opening of spectral gaps as well as effects in the localization of modes. Finally, they connect some of these ideas to properties of interest for applications, including thermoelectrics, topological insulators and photovoltaic materials.

Overall my impression of this manuscript is mixed. The paper is very well written, with the core ideas explained in a completely lucid manner. I do not doubt the technical correctness of the results, and the associated figures do an excellent job communicating the authors' key points.

Thanks!

However, the core idea itself seems underdeveloped. The question the authors are addressing is how correlated disorder is different from uncorrelated disorder with respect to electronic and vibrational states. While simulation results for minimal models are presented, only a superficial interpretation of those results is presented.

This is a valid and important point, and to address it we have now expanded our analysis and included two additional studies, summarised in a results section entitled "Generality and extension to other systems". We think these additional examples, which we discuss in more detail below, do indeed provide additional context that helps develop an improved intuitive feel for the link between correlated disorder and gap opening (in particular).

A point we would flag, however, is that 'correlated disorder' is a very broad phenomenon that encompasses many fundamentally different — if conceptually related — complex states. It is well known, for example, that the configurational entropy of these states is nontrivially dependent on the underlying geometry. Wannier's famous result for the Ising triangular antiferromagnet and Lieb's result for square ice are two of the few exact results known, and the underlying theory is sufficiently different in both cases that there is no expectation that electronic or thermodynamic phenomena should generalise to other correlated disorder states of arbitrary type.

Nevertheless we have now explored two additional model systems — one based on dimer decorations of the square lattice, and the other related to the Ising triangular antiferromagnet — that provide additional context for our study. The details are of course different to those of the icelike hidden order model of our main case study, but we do again find that strongly correlated disorder leads to qualitatively-similar results; e.g. gap opening.

For example:

- Why does correlated disorder appear to open a band gap? Is there any conceptual understanding of this key result?

We have expanded our discussion of this point, which is probably most intuitively addressed in the context of the square-lattice dimer model, for which the electronic states are most easily understood.

- Why does it do so at some high-symmetry points but not others? [e.g. in panels j,k of Fig. 1 a gap opens at Gamma, but at M. In panels n,o a large gap opens at high energy at M, but not at low energy and not at all at Gamma].

Based on the various models we have studied, we think it is not necessarily the k-points per se that are important, but the number of accumulated states at which the gap opens. The example that we discuss in detail in the text is that of the dimers on the square lattice, where the gap opens so as partition states into bonding and antibonding dimer combinations. Likewise in our A_2B structure, the key filling fraction corresponds to that required for electron-precise A_2B 'molecules'. Wherever this filling fraction cuts the electronic band structure then determines the branches that are most strongly affected by gap opening.

- Why is the gap opening locations different for phonons vs. electrons?

Similarly to the electronic case, the number of accumulated states is what determines the gap energy (and hence opening locations). These openings effectively partition the modes into different groups. In the simplest case these are the inter- and intra-'molecular' degrees of freedom. The reason there are differences to the k points of the gap opening for electronic vs phonon is due to the different band structures in the ordered cases, which originate from the different symmetries of electronic (s-p and s-s) hopping to the phonon (x-x, x-y, ...) force constants.

These points have been added to the manuscript now.

The lack of any further detail or exploration of the obtained results makes it difficult to ascertain their significance. For example, the 2-in/2-out case should have algebraic correlations between the atomic positions, while other kinds of correlated disorder may only yield exponential decaying correlations, and the "perpendicular" state has long-range correlations along each chain. Are the behavior of these correlations important to the result? Given that only a few simplified models are explored in this work, it is unclear to me how general the authors' result really is, given the absence of a conceptual framework.

We hope that our expanded our analysis and two additional studies, will have addressed the referee's key concerns here. While it is an interesting idea to explore, we have not found any evidence that the algebraic vs exponential decay of correlations is crucial to explaining the trends in band structure that we have observed.

This issue I think is my main concern with the work. While the presentation is excellent and the question posed is interesting, without a more conceptual understanding it isn't clear to me that this work should be published in its current form. My recommendation would thus be to not accept the manuscript in its current form, but perhaps to reconsider if the authors could provide a more conceptual picture for their core results.

We thank the referee for their constructive comments, which have helped us improve the manuscript.

More technically, I have a few questions/comments about more specific statements:

1) On page 7 the authors discuss thermodynamic stabilization of hidden order. They argue that the opening of a gap in the phonon or electron DOS for the "perpendicular" order suggests that a kind of Peierls distortion type transition could occur, favoring the "hidden order" state. While interesting, a natural question would be whether reducing the degeneracy further would result in an even larger gap. I.e. selecting a particular state from the manifold of "perpendicular" states could give rise to an even lower energy. If so, it is not clear to be that the partial favoring discussed by the authors is meaningful.

This is a great point which we have now addressed in the text and in a dedicated section of the supporting information. We find that there is no further electronic stabilisation by selecting an ordered state from amongst the disordered manifold — at least not without considering coupling to strain. Consequently, for systems with extensive configurational degeneracy of the hidden

order state (e.g. dimer formation in pyrochlores) the disordered Peierls state will always be favoured at finite temperature.

2) While the spectral plots shown in Figs. 1,3 seem satisfactorily converged, the DOS presented in Fig. 2 remains very noisy. In the supplemental information, the authors state 30 random samples were used. This is not nearly enough to obtain a converged DOS, as can be seen in Fig. 2. From this figure the difference between the random and correlated disorder cases seems well within the noise and thus it isn't clear what kind of conclusions can really be drawn.

We agree that these figures were not entirely satisfactory, and have improved the calculations in terms of system size and number of configurations. We now use systems of 60 by 60 atoms and average over 100 configurations. We feel that these results have become much clearer now.

3) When discussing the degree of localization of the modes, can the metric used distinguish between 0D and 1D localization? Given the "chain" like structure of the perpendicular states, it seems possible they could be localized in one direction but not the other. I'd imagine this could be diagnosed by looking at scaling in system size. Can the authors comment on this?

We have found it easiest to understand localisation through direct interpretation of the eigenvectors, as presented here, but the referee is right to flag that scaling with system size might be an alternative metric.

Reviewer 2

This manuscript describes a theoretical demonstration of the influence of correlated disorder on electronic and phononic properties. The paper is well written, and the results are clear and interesting.

Thanks!

I have one quite general criticism and then a number of specific comments that the authors should consider. Once addressed, I feel this manuscript would be appropriate for publication in Nature Communications.

Thanks!

My general criticism is that the authors refer to the correlated disorder that they model and analyse, as “hidden order”. I can understand why they do this, as such incomplete order does not show up in conventional diffraction studies, and they make the case that it is often overlooked (which I am sure is the case). However, it isn’t really “hidden”. As the authors show, the different manifestations of correlated disorder show quite distinct diffuse scattering. While this is a criticism, I would be fine with it. However, “hidden order” is a well-used and well known term in correlated electron physics, and typically refers to thermodynamic phase transitions which have no easily-interpretable signature in structural probes, such as neutron and x-ray scattering. The best appreciated specific example is that of the heavy fermion superconductor URu₂Si₂, which the authors refer to, and which displays a very strong thermodynamic anomaly at ~ 17 K in zero field and ambient pressure. There have been many unsuccessful attempts to identify weak Bragg peaks associated with this thermodynamic anomaly, and hence the state below ~ 17 K is referred to as a “hidden order” state. There have been many proposals and speculations as to the nature of this “hidden order” state, but all the ones which I am familiar with propose exotic electronic ordering, such as high order multipolar order consistent with 5f-electron physics, and none propose a structural correlated disorder origin.

So, at least some of the community will be confused by the use of “hidden order” to describe the correlated disorder that is the subject of this manuscript. That is unless the authors are suggesting that correlated disorder is relevant for URu₂Si₂. Were that the case, among other things, the authors would need to discuss why a thermodynamic phase transition is required.

For this reason, I strongly recommend that the authors either simply refer to the “hidden order” as correlated disorder, or provide another descriptive term, or better explain the relationship between the “hidden order” described here and that commonly used in the correlated electron community.

Thank you for raising this point, which is particularly important given the broad audience of Nature Communications. We certainly do not intend to imply that correlated structural disorder is relevant to URu₂Si₂, and have now included an additional note in the text to clarify our use of the term “hidden order”. We have also removed a previous mention of URu₂Si₂ to better avoid any confusion about the use of the term. The term is increasingly being used beyond the correlated electron physics community, e.g. by Paul Attfield in his description of orbital-molecule phases, to describe states with well-defined local order that are not long-range ordered. Hence, as for the phase transition URu₂Si₂, there is no signature of this order in the Bragg contribution to neutron and x-ray scattering patterns, nor any symmetry breaking. We have included a figure in the supporting information (Figure S1) and accompanying text to show how subtle is the diffuse scattering contribution associated with this hidden order — consistent with the vernacular use of “hidden” to mean something present, but difficult to see.

To address points raised by other referees, we have now included some discussion of additional models, and would point out briefly that the square lattice dimer model we consider (Fig 5a) contains no structural disorder; here the hidden order is purely electronic in that it involves a correlated disorder in hopping probabilities.

My more minor comments are:

1- A very minor point: when I look at the 2D structures in Fig. 1 a) – d) I immediately think of CuO₂ planes, as occurs in many quasi-2D high temperature superconductors. Would this be inappropriate? I imagine H₃S is as good or perhaps a better conceptual analogy to use.

This is a nice analogy, but one we would be hesitant to explore in detail in the current manuscript. The valence orbital symmetries are simpler for H₃S than for CuO₂, which is the key motivation for our use of the former rather than the latter. We have added short mention of the relation of the system to cuprates.

2- As the calculated electron and phonon bands span the 2D Brillouin zone shown in Fig. 1 and 3, along high symmetry directions involving the X and M points. It may be useful to provide a drawing of the 2D Brillouin zone indicating these points, so that a less expert reader could follow this more easily.

This is a good point, and we have now included in the legend an explicit description of the coordinates of the key high-symmetry points to help the reader.

3- The measures of electron and phonon delocalization shown in Fig. 3 are interesting, however, how these measures are determined and presumably related to the electronic and phononic band structures shown in Fig. 1 requires that the reader continue into the Methods section. Given the prominence of this result – accounting for 1 of the 4 figures in the manuscript, I would prefer to have these details discussed, at least qualitatively, where Fig. 3 itself.

This is a good point and we have now added further detail regarding these calculations to the main text as requested.

Reviewer 3

This paper presents simulations that demonstrate how varying local order types within a toy model average crystal structure can give rise to different features within electronic and phononic band structures, specifically blurring of bands and opening of new gaps. The work is from a leading group in the field of hidden order within crystals and the calculations appear technically sound. Graphical comparison of results throughout is excellent.

Thanks!

However, glossing over the electron-filling aspect of the electronic band structure is a weakness. Even in a toy model, electronic structural stability is implicit for the pDOS to be meaningful and so some number of electrons (such as for H₂O) should be used to define a proper Fermi level. The eDOS's for the ordered and random models in Fig. 2a look rather similar with the implication that both would be metallic for EF somewhere in the middle of the band which seems odd.

This is a good point, and we have now included additional discussion at various points regarding the relevance of band gap formation to electron count (note also our response to referee 1 above). Figures 1 and 5 now contain explicit markings for the Fermi level. Figure 2c also addresses the point of electronic stability.

The eDOSs should also be sensitive to the magnitude of displacement of A from the midway position used in the calculations. Was this varied and does it have any significant effect?

The referee is right of course that the magnitude of A displacement affects the eDOSs, effectively by changing the hopping amplitudes, but the relationship is straightforward in that the variation from the ordered eDOS is continuous and monotonic as displacement magnitude is varied. Hence there is no conceptually significant effect, which is why we have not commented on this point when discussing the A2B system in the text. With the addition of the square dimer system in fig. 5 we now show more explicitly an example of how the hopping amplitudes changes gaps.

The issue of electron-electron correlation which opens Mott type gaps in many materials is also not mentioned - in real systems it may be hard to distinguish this from disorder-induced gaps.

This is a good point. Our model intentionally excludes Mott physics, which is what allows us to conclude that electron-electron correlation may not necessarily need to be invoked to rationalise gap opening in certain systems. We now include a comment to this effect in the main text. Obviously our model is also different to the extreme disorder at the heart of Anderson physics.

The main novelty of the paper seems to be as an educational exercise in being able to eyeball how the different types of real space disorder translate into momentum space features of diffuse scatter and band structure changes. As such I found it very interesting and I think it will have impact on others working on disordered materials.

Thank you.

The downside is that I'm not sure if any of the calculated features will be observable. People have been studying hidden disorder materials such as those refs 9-24 and also using supercell models to simulate band structures of disordered materials for many years but I do not recall seeing specific examples where a disorder model is linked to specific changes in electron/phonon structure. There is the long-known link of disorder to metal-insulator transitions, going back to Anderson localisation theory, but I'm not sure the present work adds much to this, perhaps because electron-electron correlation is neglected as above. Fig. 2a,b, shows some large gaps being opened in some models - are these reported for the corresponding materials or why were they not found?

As discussed above, the model we present here is conceptually distinct from Anderson theory, albeit that both explore a link between disorder (of different kinds) and electronic structure. We have now included in the main text two additional models, and accompanying discussion that highlights links between the toy-model results presented here and the established behaviour of

some key experimental systems. We think in particular that the electronic description of disordered orbital-molecule phases is closely related to the picture that emerges from our study.

The Concluding Remarks give some suggested connections to topical materials like topological insulators, but there is a lack of specific predictions. How would I use this paper to do some new disorder calculation or study that I couldn't already do if I was working on TI's or other materials?

This a good point and we have now included some additional discussion along these lines where we point out some important groups of TI materials where we expect this to have an effect.

Overall I find this a stimulating paper that could be accepted for publication after consideration of the above comments.

Thanks!

REVIEWERS' COMMENTS

Reviewer #1 (Remarks to the Author):

In response to the reports of the referees the authors have substantially modified their manuscript, adding a handful of new and interesting examples to explore the core ideas of their work.

These additions directly address what I had considered the key shortcoming of the original paper and I think improve the work considerably. The authors have also address my more specific comments satisfactorily and (in my opinion) have addressed the majority of the issues flagged by the other referees.

Given these updates I have no reservations in recommending publication in Nature Communications.

Reviewer #2 (Remarks to the Author):

The authors have responding adequately to my specific queries and criticisms. I have read through their response to the other two referee reports, and feel that these criticisms have also been seriously addressed. As my original recommendation was a provisional acceptance, I am happy to recommend acceptance of this manuscript in Nature Communications.

Reviewer #3 (Remarks to the Author):

The authors have responded extremely well to the reviewer queries and suggestions. In regard to my own previous comments, I note that band filling is now covered, and suggestions of how this approach might be useful to some real materials such as oxide spinels and perovskites, as well as specific topological insulator materials, are now given.

As before, I feel that this is a very stimulating and insightful paper that will be useful to those working on disordered but locally correlated materials, and I now recommend acceptance for publication as is.

RESPONSE TO REVIEWERS' COMMENTS

Reviewer #1 (Remarks to the Author):

In response to the reports of the referees the authors have substantially modified their manuscript, adding a handful of new and interesting examples to explore the core ideas of their work.

These additions directly address what I had considered the key shortcoming of the original paper and I think improve the work considerably. The authors have also address my more specific comments satisfactorily and (in my opinion) have addressed the majority of the issues flagged by the other referees.

Given these updates I have no reservations in recommending publication in Nature Communications.

Reviewer #2 (Remarks to the Author):

The authors have responding adequately to my specific queries and criticisms. I have read through their response to the other two referee reports, and feel that these criticisms have also been seriously addressed. As my original recommendation was a provisional acceptance, I am happy to recommend acceptance of this manuscript in Nature Communications.

Reviewer #3 (Remarks to the Author):

The authors have responded extremely well to the reviewer queries and suggestions. In regard to my own previous comments, I note that band filling is now covered, and suggestions of how this approach might be useful to some real materials such as oxide spinels and perovskites, as well as specific topological insulator materials, are now given.

As before, I feel that this is a very stimulating and insightful paper that will be useful to those working on disordered but locally correlated materials, and I now recommend acceptance for publication as is.

We would like to once again thank the referees for their time and comments, which helped improve the manuscript.